# Experimental Study on the Effects of L(+)-Ascorbic Acid Treatment to the ESR Signals of Human Fingernails

Chryzel Angelica B. Gonzales [1,*], Ryogo Ajimura [2] and Hiroshi Yasuda [1]

1   Research Institute for Radiation Biology and Medicine, Hiroshima University, 1-2-3 Kasumi, Minami-ku, Hiroshima City 734-8553, Hiroshima, Japan
2   School of Medicine, Hiroshima University, 1-2-3 Kasumi, Minami-ku, Hiroshima City 734-8553, Hiroshima, Japan
*   Correspondence: chryzelgonzales@hiroshima-u.ac.jp; Tel.: +81-82-257-5872

**Featured Application: The L(+)-ascorbic acid treatment prior to radiation dosimetry using fingernails could efficiently remove the disturbing ESR signals caused by heat or UV light exposure.**

**Abstract:** The effects of L(+)-ascorbic acid (AA) (an antioxidant commonly known as vitamin C) on the electron spin resonance (ESR) signals from fingernails were examined in relation to X-ray and UV irradiation. The ESR signal intensity, stability, and sensitivity to radiation were measured under different storage conditions. The experimental results indicated that the behavior of the increase in the ESR intensity of the AA samples varied depending on the storage and location conditions, showing sensitivity to light and signal instability at room temperature. It was found that the AA treatment caused a large increase in the peak-to-peak intensities with continuous signal growth with storage time, which may provide an enhancement to the radiation-dependent signal in fingernails. It was also suggested that the use of AA for pre-treatment could sufficiently remove the disturbing signals induced by heat or UV light exposure, which is expected to improve the reliability of radiation dosimetry using fingernails. Further studies with different antioxidant conditions are needed to better characterize the complex changes of the ESR signals from fingernails.

**Keywords:** radiological accident; retrospective dosimetry; ESR; EPR; fingernails; antioxidants; ascorbic acid

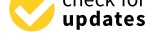



## 1. Introduction

The electron spin resonance (ESR; commonly called electron paramagnetic resonance (EPR)) from human fingernails has been of special interest in radiation accident dosimetry for their ubiquity and convenient sampling. The exposure of fingernails (and other keratinous tissues) to ionizing radiation generates free radicals, which are the primary sources of the ESR signals. The typical ESR signals in fingernails are categorized into three types: the radiation-induced signal (RIS; produced by exposure to ionizing radiation), mechanic-induced signal (MIS; generated by mechanical stress associated with cutting/clipping), and background signal (BGS; an omnipresent non-radiation-induced signal). In recent years there have been a few reported potentially confounding factors that could affect neither the intensity (amplitude) nor the spectrum shape of the BGS and RIS. These factors include the direct exposure to light with ultraviolet (UV) component [1,2], which was interpreted as the light-induced signal (LIS), and the thermal effect caused by high temperature [3,4]. An observed strong effect of these factors can be seen in the visual shape changes of the BGS spectrum, which can, in turn, affect the RIS intensity in irradiated fingernails.

While the difficulty to remove (or reduce) the presence of the highly variable BGS from the ESR spectrum is not particularly surprising, different approaches have been studied to overcome this problem, for example, the treatment of the samples using water [5–7] or chemical like dithiothreitol (DTT) reducing agent [8–10]. However, it is difficult to

provide a clear view of the positive influence of these treatments because not only BGS can be reduced in the spectrum but also the RIS intensity. Moreover, the use of DTT reducing agent for chemical treatment may provide toxicological effects when handling is not properly managed. With these challenges concerning the BGS, an effort to identify its possible nature or origin has been given attention in a more recent study [11]. The authors of the study advanced that common radicals (i.e., o-semiquinone) are responsible for the RIS and BGS: the RIS being produced from exposure to ionizing radiation and the BGS from UV effects, and thus both signals are characterized to be virtually and chemically identical [11]. This hypothesis has led to a suggestion that chemical treatment methods using oxidizing (0.1 M potassium ferricyanide) or reducing (0.1 M DTT) agents are unlikely to yield the potentials to discriminate the RIS and BGS [11]. On the other hand, given the limited published research literature to date, further studies are expected to investigate the interactions of various chemicals such as antioxidants on quenching free radicals. Moreover, as UV exposure has the effect of generating a strong ESR signal [1] and that the BGS is surmised to be produced by the UV effects [11], it is also a good opportunity to test the use of such antioxidants to reduce UV-related exposures which could occur during daily activities but are not well considered yet.

The aim of our study was to investigate the application of a non-hazardous chemical that can work as a natural antioxidant and scavenger of free radicals. The current work focused on the use of L(+)-ascorbic acid (hereafter "AA"), known as vitamin C, and examined the effects of AA treatment on the ESR signals of unirradiated and irradiated fingernails (i.e., X-ray and UV irradiation). It is anticipated that the AA may react vigorously to ambient conditions and, therefore, the influence of AA on the ESR signals of fingernails was demonstrated using two different storage mediums (vacuum and non-vacuum) under a variety of location conditions. The effects of the AA treatment were analyzed based on the signal stability and sensitivity change before and after the treatment or irradiation.

## 2. Materials and Methods

All fingernail samples were obtained from one volunteer (female, age 31) and collected within a two-week time interval. The collected samples were placed in a commercially available sealable packet and stored inside a built-in vacuum chamber (VE-ALL, AS ONE Corporation, Osaka, Japan) prior to the actual experiments. In the present study, we used a total of nine sample collections for the two experimental procedures described below (see Figure 2). In Experiments 1 and 2, the corresponding number of utilized sample collections were 4 and 5, respectively. In each collection, the cut samples weighed a total of ~90 mg, with each aliquot's length and width in the range 7–10 mm and 1.5–2 mm, respectively. We divided the cut samples from each collection into four subsets: three for AA treatment and one for water treatment (hereafter 'WT') as (background) control. Each subset was composed of 5–10 aliquots weighing 20–21 mg.

The synthesis of the treatment solution was made by dissolving 1.06 g of AA (FUJI-FILM Wako Pure Chemical Corporation, Osaka, Japan) in a 60 mL Milli-QTM ultrapure water (resistivity = 18.2 M $\Omega$ cm) using a digital magnetic stirrer (MS-H-PROT, DLAB Scientific Inc., California, United States) at 200 rpm for 5 min. The resulting treatment solution is equivalent to 0.1 M concentration of the AA. In procedures with AA, each of the three subsets was treated with 500 µL AA solution, whereas the remaining one subset was only treated with 500 µL of water (for control). The treatment time for all four subsets was set to 5 min. After the corresponding treatment, all treated samples were dried for 1 h inside a vacuum container (ZWLLING J.A. HENCKELS, Solingen, Germany) with the lid sealed using the designated vacuum pump (ZWLLING J.A. HENCKELS, Solingen, Germany). An illustration of the treatment procedure followed by drying is shown in Figure 1.

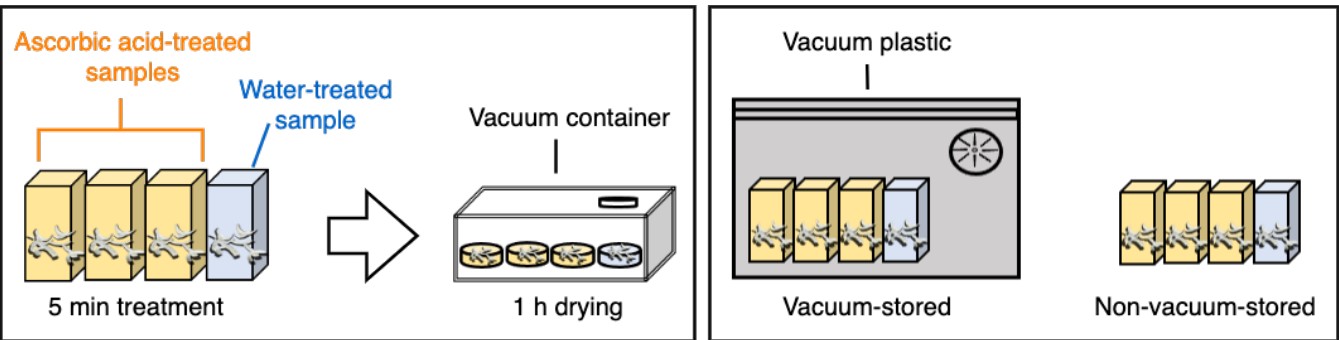

**Figure 1.** Example illustrations of the treatment procedure of the samples followed by drying inside a vacuum container (**left**) and the storage procedure in two different mediums: vacuum and non-vacuum (**right**).

**EXPERIMENT 1: AAT1 ⇨ IR1 ⇨ AAT2 ⇨ IR2**

| STEP 1: AAT1 | ESR measurement (Initial signal) | Ascorbic acid treatment 1 (500 µL, 5 min) | Drying (Vacuum, 1 h) | ESR measurement (Soon after drying) |

| STEP 2: IR1 | ESR measurement (Before IR1/ soon after drying) | Irradiation 1 (X-ray, 20 Gy) | ESR measurement (Soon after IR1) | ESR post-measurement (up to 7 days) |

| STEP 3: AAT2 | ESR measurement (7 days after IR1) | Ascorbic acid treatment 2 (500 µL, 5 min) | Drying (Vacuum, 1 h) | ESR measurement (Soon after drying) |

| STEP 4: IR2 | ESR measurement (Before IR2/ soon after drying) | Irradiation 2 (X-ray, 20 Gy) | ESR measurement (Soon after IR2) | ESR post-measurement (up to 7 days) |

**EXPERIMENT 2: UV ⇨ AAT1 ⇨ IR ⇨ AAT2**

| STEP 1: UV | ESR measurement (Initial signal) | UV exposure (30 min) | ESR measurement (Soon after UV exposure) | ESR post-measurement (up to 7 days) |

| STEP 2: AAT1 | Ascorbic acid treatment 1 (500 µL, 5 min) | Drying (Vacuum, 1 h) | ESR measurement (Soon after drying) | ESR post-measurement (up to 7 days) |

| STEP 3: IR | ESR measurement (7 days after AAT1) | Irradiation (X-ray, 20 Gy) | ESR measurement (Soon after IR) | ESR post-measurement (up to 1 day) |

| STEP 4: AAT2 | Ascorbic acid treatment 2 (500 µL, 5 min) | Drying (Vacuum, 1 h) | ESR measurement (Soon after drying) | ESR post-measurement (up to 9 days) |

**Figure 2.** Flow diagram for the specific steps used for Experiments 1 (**above**) and 2 (**below**). The notations UV, AAT, and IR represent ultraviolet, L(+)-ascorbic acid treatment, and irradiation, respectively.

The irradiation (hereafter "IR") of the samples was conducted using a 160 kVp X-ray beam (6.3 mA) (Cabinet X-ray System Model 43855F with CP160 Option, Faxitron X-ray LLC, Illinois, United States). The system consists of an X-ray tube (MXR-160/22, Comet AG, Flamatt, Switzerland) with 0.8 mm Be inherent filtration. The source-to-surface distance was set to 23 cm. The dose rate was 2.71 Gy/min. Each of the subset samples was exposed to a fixed total dose of 20 Gy. The samples were positioned between the 0.5 cm ($\varphi$ 8 $\times$ 8 cm) and 0.1 cm ($\varphi$ 10 $\times$ 10 cm) thick solid water-equivalent phantom plates (PH-40 Tough Water Phantom®, Kyoto Kagaku Co. Ltd., Kyoto, Japan). The irradiation setup is illustrated in Figure 3.

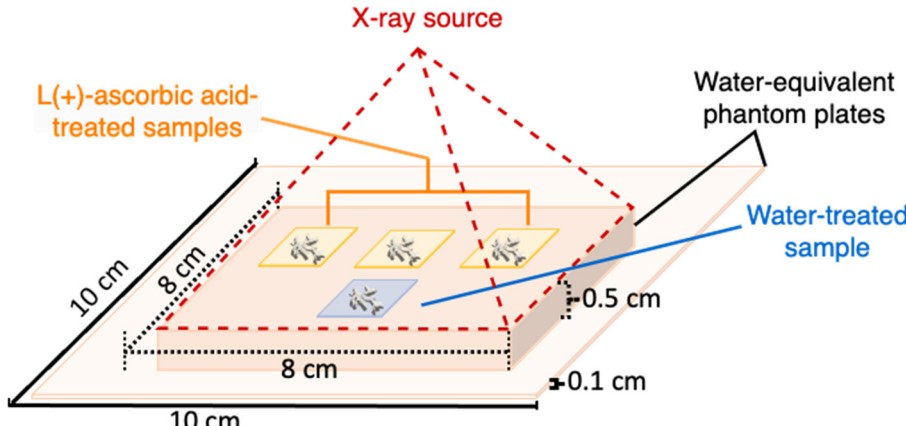

**Figure 3.** Schematic of the irradiation setup for the fingernail samples exposed to 20 Gy of X-rays.

Between UV exposure, treatment, irradiation, or measurement procedures, the samples were stored in two different mediums (see also Figure 1): (a) vacuum storage, inside an easy-zip vacuum bag (ZWLLING J.A. HENCKELS, Solingen, Germany) with the valve sealed using the designated vacuum pump (ZWLLING J.A. HENCKELS, Solingen, Germany) and (b) non-vacuum storage. Both vacuum-stored and non-vacuum-stored samples were placed in various location conditions (for example, 20 °C or −20 °C temperature; with or without exposure to ambient light). Specific details of the corresponding storage and location conditions for the two experimental procedures are shown in Table 1. These various storage and location conditions allowed us to observe the changes in the ESR signals of the AA-treated samples.

**Table 1.** List of various experimental storage and location conditions used for the fingernail samples in Experiments 1 and 2.

| Experiment | Without Light Exposure (in Darkness) | | | | With Light Exposure (in Ambient Illumination) | | | |
|---|---|---|---|---|---|---|---|---|
| | Vacuum-Stored | | Non-Vacuum-Stored | | Vacuum-Stored | | Non-Vacuum-Stored | |
| | RT | LT | RT | LT | RT | LT | RT | LT |
| 1 | ● | ● | - | ● | ● | - | - | - |
| 2 | ● | ● | ● | - | ● | - | ● | - |

Notes: ●: used; -: not used; RT: room temperature (20 °C); LT: low temperature (−20 °C).

ESR measurements were performed at room temperature on an X-band spectrometer (JES-FA100, JEOL Ltd., Tokyo, Japan) operating at ~9.441 GHz with an ES-UCX2 resonator using a 5-mm diameter sample tube. Parameters used for the spectra acquisition were: 1 mW microwave power; 0.4 mT modulation amplitude; 10 mT sweep width; 30 s sweep time; 0.03 s time constant; the number of scans was 10. The corresponding nominal values of the third and fourth line of the MgO:Mn$^{2+}$ reference sample at g = 2.033 and g = 1.981 were used for the g-factor adjustment. ESR measurements were conducted at different

elapsed times from 0 (immediately) up to 7 (or 9) days after UV exposure, treatment, or irradiation procedure. Throughout this paper, all recorded spectra were normalized to their respective sample weight. The ESR spectra from the three AA-treated subset samples were averaged and subtracted from the water-treated (background) control sample. The resulting background-subtracted signal was displayed as a single spectrum or datum point in the plots. The reported error values were calculated as the standard deviation of the three AA-treated subsets of samples. The temperature and humidity during measurements were always monitored using a dual channel data logger (LR5001, Hioki, Nagano, Japan) with a built-in sensor (LR9504, Hioki, Nagano, Japan) and a communication adapter (LR5091, Hioki, Nagano, Japan).

In the experiment with UV exposure, the samples were exposed for 30 min at room temperature using a portable UV light emitting device with a luminance intensity of 170 mW/cm$^2$ at 365 nm wavelength (Portable Cure 100, HLR100T-2, SEN LIGHTS Corporation, Osaka, Japan). Note that the portable UV light was situated at room temperature laboratory condition.

The experimental procedures used in the present study were designed to investigate the effects of AA treatment (hereafter "AAT") (a non-toxic antioxidant) on the ESR signals of the irradiated (i.e., X-ray and UV irradiation) and unirradiated fingernails. Repeated treatments and irradiations were done to examine the behavior changes of the ESR peak-to-peak intensities in terms of signal stability and sensitivity to radiation. All procedures (except during irradiation) were conducted under subdued red light.

## 3. Results and Discussion

Figure 4 illustrates the peak-to-peak intensities obtained from four different sample sets following repeated treatment and irradiation procedures, as described in steps 1–4 of Experiment 1. All sample sets were recorded in the same measurement times following the first treatment (AAT1 or WT1) and 20 Gy X-ray irradiation (IR1) procedures. Each sample set was allocated for a specific storage and location condition (in this case, four sample sets mean four different storage and location conditions). As mentioned previously, each collection was divided into four subsets: three for AAT and one for WT (control). The displayed data for the AAT samples were obtained as the average of peak-to-peak intensities (or amplitudes) from three AAT subsets and background-subtracted from the WT (control) subset, as indicated by the unfilled bar in Figure 4. Data on the WT subset from each four sample collections were also included in the plot for comparison, as denoted by the patterned bar, also presented in Figure 4. Figure 4A,B were vacuum-stored samples placed at room temperature (RT, 20 °C) without light exposure (in darkness) and with light exposure (in ambient illumination), respectively. Figure 4C,D, on the other hand, were vacuum-stored and non-vacuum-stored samples, respectively, and both were placed at low temperature (LT, −20 °C) without light exposure. It should be noted that the experimental steps and measurements for the "soon after AAT1/WT1" and "soon after IR1" in Figure 4A–D were performed on the same day for all the samples. In this situation, the peak-to-peak intensities should be the same for all four sample collections (in both AAT and WT samples) regardless of the indicated storage and location conditions; the storage procedure only started after the end of the first measurement following the first irradiation (labeled as "soon after IR1"). Here the results of the "soon after AAT1/WT1" and "soon after IR1" were seen to be consistent in both AAT and WT samples. One interesting observation were the subsequent dose-response of the AAT samples provided plausible reproducibility with low intra-individual variations. The probable reason for this may be related to the radiation-specific radical responsible for the generation of the radiation-induced signal in fingernails, which might not be affected by the AAT prior to X-ray irradiation. This important finding indicated a good starting value for a reliable observation on the effects of AAT on the ESR signals and the behavior changes under various storage and location conditions.

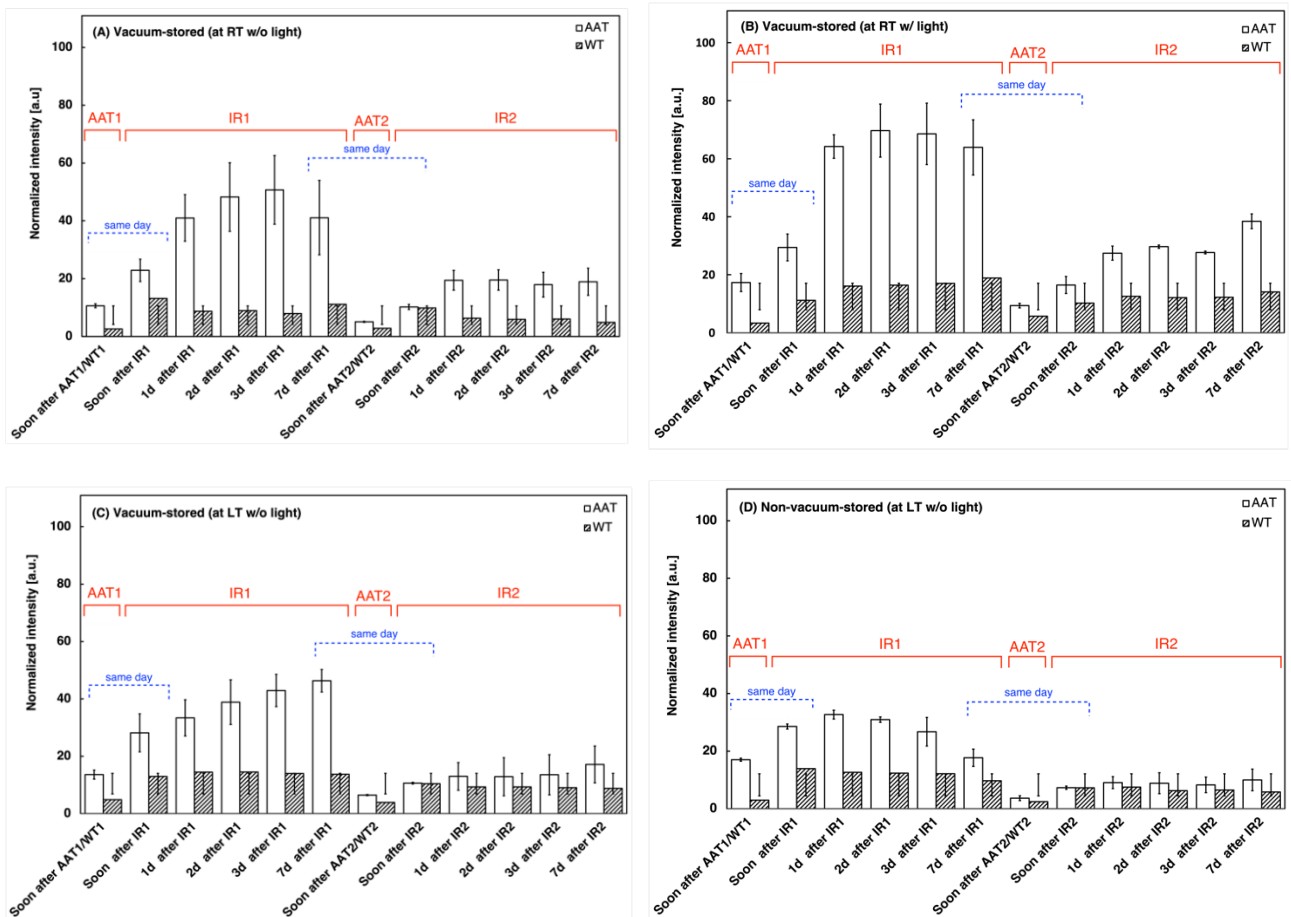

**Figure 4.** Comparison of the effects of repeated L(+)-ascorbic acid (AAT) or water treatments (WT) and X-ray irradiations (IR) to the ESR peak-to-peak intensities in samples stored under four different location conditions, as described in steps 1–4 of Experiment 1. Each point of the AAT samples in the plot was calculated as the average of three samples and subtracted from the WT control sample. The error bars on the AAT samples were obtained as the standard deviation of three samples, while one standard deviation was used for the WT sample. Note that "soon after AAT1" and "soon after IR1" were performed on the same day, as well as the "7d after IR1", "soon after AAT2", and "soon after IR2". The notations RT and LT correspond to room temperature (20 °C) and low temperature (−20 °C), respectively. The with (w/) and without (w/o) light imply that the samples were placed in ambient illumination and in darkness, respectively.

Following post-irradiation measurements (up to 7 days) after the "AAT1-IR1" process (steps 1–2 of Experiment 1), a large increase in the ESR signal with storage time was observed in all samples with the increase rate depending on the storage and location conditions. The signal intensity of the vacuum-stored AAT samples placed at LT without light exposure increased steadily (from 1 to 7 days) (see Figure 4C), whereas the other vacuum-stored AAT samples placed in RT condition (both with and without light exposure) initially experienced an intensity increase (from 1 to 3 days) with a subsequent decrease observed possibly due to the signal fading of the RIS in RT condition (see Figure 4A,B). The same fading behavior was also observed in the non-vacuum-stored AAT samples placed at LT with no light exposure (see Figure 4D); the fading effect started much sooner but slower following irradiation. Another observation was the considerable difference in the signal growth between the AAT and WT samples. This signal growth of the signal intensity in AAT samples varied depending on the storage and location conditions. The largest signal increase was observed in the vacuum-stored AAT samples placed at RT with light exposure (see Figure 4B) in comparison to those without light exposure (refer to Figure 4A,C,D),

which may be related to the sensitivity of fingernails to ambient illumination [1,2]. Generally, the WT samples, on the other hand, were found to be stable with no drastic signal growth under vacuum storage, but the combination with the LT condition was able to achieve a more stable signal with very minimal (almost zero) signal changes.

For completeness, the second treatment (AAT2 or WT2) and irradiation (IR2) procedures (steps 3–4 of Experiment 1) were applied in the same way and in the same sets of samples, also shown in Figure 4. The radiation sensitivity of the AAT samples changed with repeated treatment and irradiation. The absolute signal intensities were significantly reduced (>50%) for all AAT samples. Although it can also be noticed that the increase rate of the signal regrowth was slower with improved signal stability in comparison to the first treatment and irradiation, the reduction of sensitivity to radiation still possesses a major disadvantage. The signal intensities of the vacuum-stored AAT samples at RT with light exposure also continued to increase during storage time. In contrast, the continuous signal regrowth with post-irradiation storage time and concomitant changes to the radiation sensitivity was not observed in the WT samples after repeated treatment and exposure to radiation, as also shown in Figure 4.

To summarize the obtained results in Experiment 1, we found that all the samples following AAT exhibited an increase in their signal intensities. This increase in the ESR peak-to-peak intensity suggests that the use of AA for sample treatment may provide an enhancement effect to the radiation-dependent signal of interest needed for reliable ESR fingernail dosimetry. While the AA was also observed to be very unstable at ambient conditions (i.e., temperature, humidity, and light), the enhanced intensity due to the signal generated by the AA treatment was significantly dependent upon the storage conditions of the samples. With such well-controlled storage conditions, the degree of variability in the enhancement effect caused by the AA can be used to estimate the absorbed dose in fingernails.

According to the experimental results obtained in Experiment 1, the growth and stability of the ESR signal intensity in AAT samples were dependent upon the storage and location conditions, with the most variable dependence observed in samples with light exposure. To understand the behavior of the signal growth in AAT samples that were exposed to light with a UV component, we considered initially examining the effect of UV exposure in unirradiated and untreated fingernail samples. Figure 5 shows the time evolution of the ESR signals obtained from five different sample sets following exposure to UV for 30 min, as described in step 1 of Experiment 2 (see Figure 2). Note that each sample set was exposed to UV separately and stored under various storage conditions. As can be seen, an increase in the ESR signal was observed immediately after UV exposure (solid red line). The increase of the signal following UV exposure was about 1–2 orders of magnitude higher than the initial signal before UV exposure (solid gray line). Differences in the maximum UV-induced signals (main singlet) can also be noticed among the five sample sets, which may be due to the intra-sample variability in the fingernails' sensitivity to UV. The fading characteristics of the UV-induced signal also appeared to be variable among different storage and location conditions.

Another important finding following UV exposure was the appearance of the "bulge" in the spectra (between g = 2.014 and 2.024), which was described as the resulting effect of the fingernails' exposure to high-heating temperatures [3]. This unexpected presence of the "bulge" may be due to the temperature change during the UV exposure process. To determine the reason behind this unexpected result, we recorded the temperature and humidity in real time during the UV exposure process with the use of three different dual channel data loggers. The dual data channel loggers were placed under the UV light for 30 min for five consecutive times (this is similar to the procedure conducted during exposure of the samples to UV). The results are shown in Figure 6. Here one can see the gradual changes in the temperature and humidity with UV exposure time. The temperature and humidity continuously increased and decreased, respectively, until the UV exposure time stopped. These changes in the temperature and humidity during UV exposure showed qualitatively the same behavior, as seen in Figure 6. The average

maximum recorded temperature was 50 °C, and the average minimum recorded humidity was 8%. The increase in the temperature, which occurred continuously during UV exposure, was the reason for the apparent appearance of the "bulge" [3]. While the "bulge" was reportedly observed mostly in high-heating temperatures (i.e., beyond >80 °C), it may be possible that the UV exposure contributed to the exacerbation of its occurrence. Lastly, it should also be noted that the stability of the "bulge" behaved in a similar fashion as the UV-induced signal (main singlet) under the same storage and location conditions. In other words, the vacuum storage at LT with no light exposure was able to keep both the "bulge" (assuming this is a by-product of the heat-related induced signal) and UV-induced signal intact up to 7 days (compared to the non-vacuum storage at RT with light exposure), as illustrated in Figure 5.

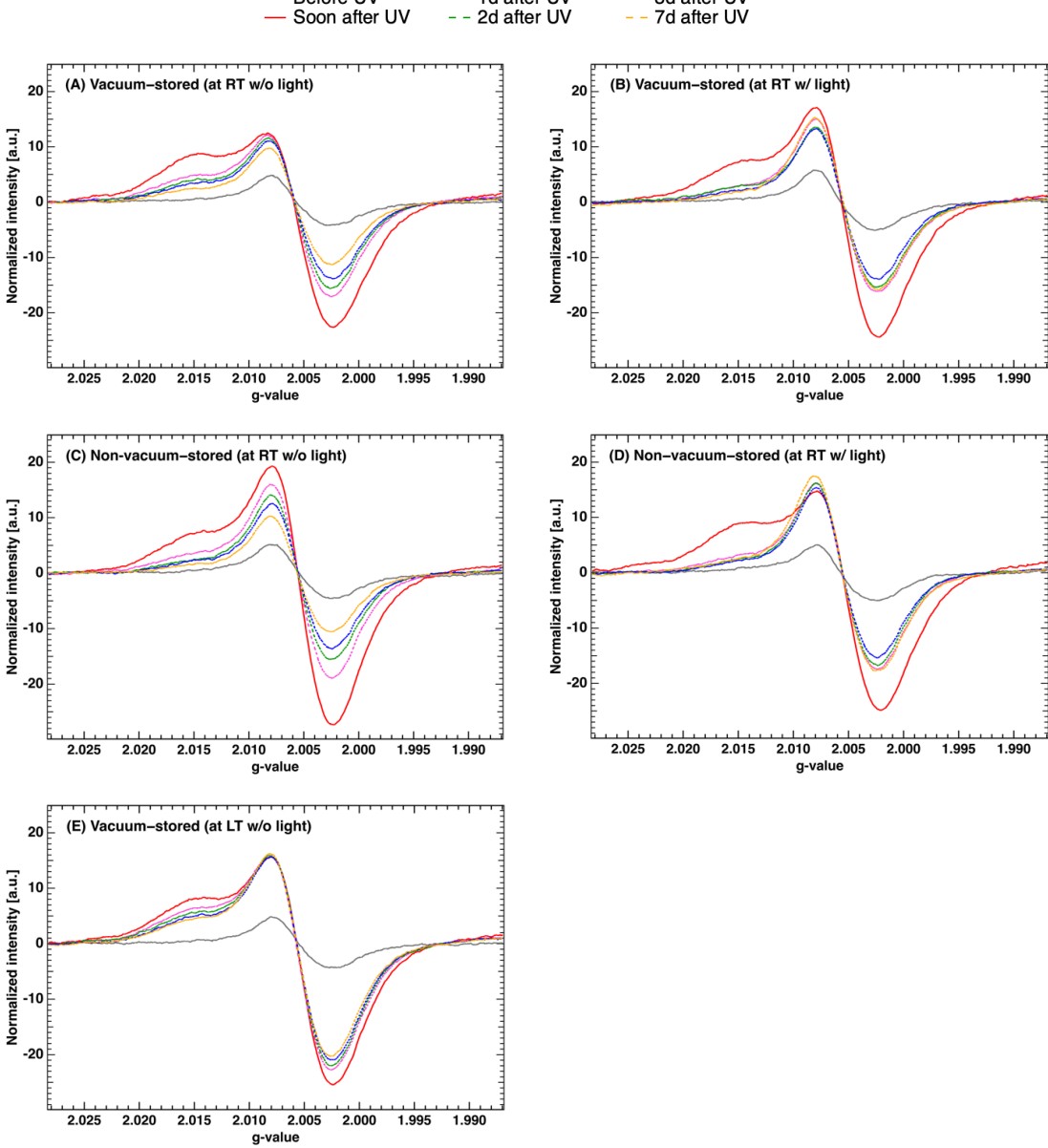

**Figure 5.** Time evolution of the ESR spectra in unirradiated and untreated samples following UV exposure for 30 min, as described in step 1 of Experiment 2. Each sample set was stored under different storage and location conditions. Each spectrum or datum point was obtained as the average of four subsets. The notations RT, LT, w/light, and w/o light are the same as in Figure 4.

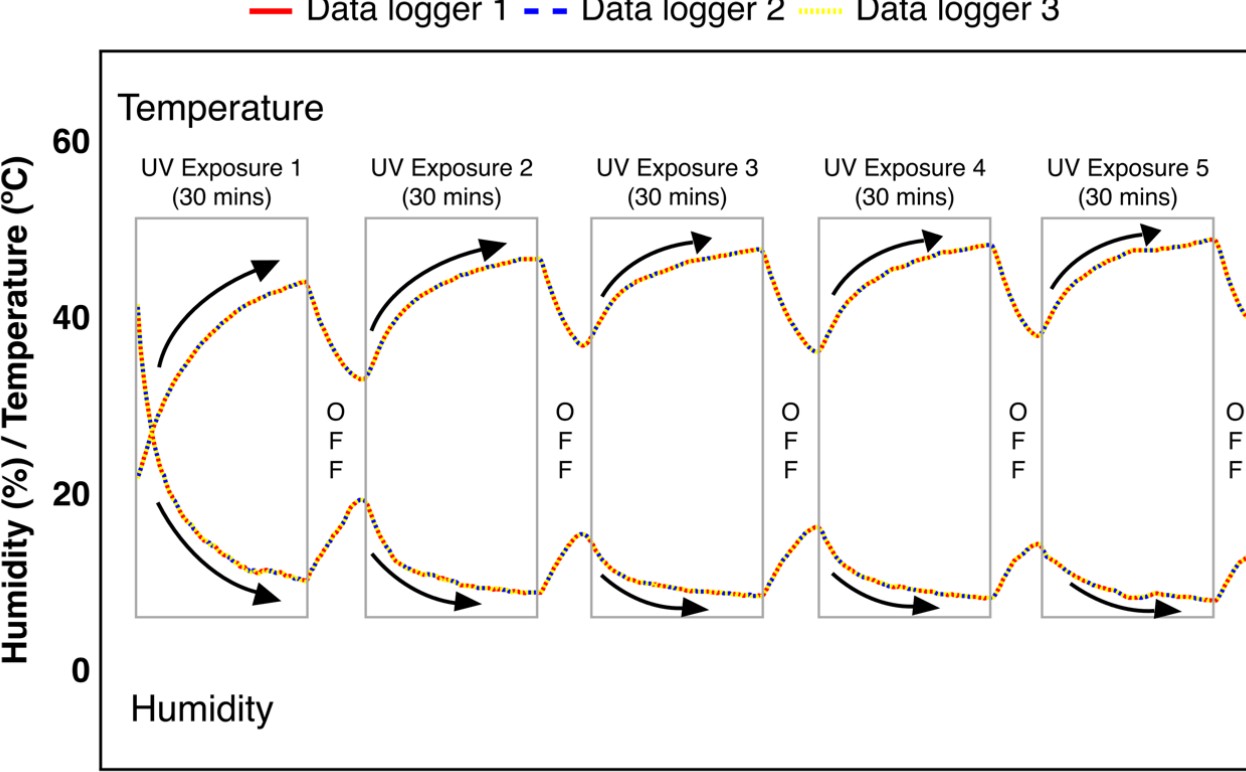

**Figure 6.** Continuous monitoring of the temperature and humidity during UV exposure recorded from three different dual data channel loggers. Each exposure was set to a power-on time of 30 min with a power-off interval of 5 min. The average maximum recorded temperature was 50 °C, and the average minimum recorded humidity was 8%.

Figure 7, on the other hand, shows the comparison of the peak-to-peak intensities obtained from five different sample sets stored under various storage conditions, as described in steps 1–4 of Experiment 2. Plotted values from AAT1 (i.e., starting from step 2) were calculated as the average of three AAT samples and subtracted from the (background) control sample. Note that the plotted peak-to-peak intensity values for the UV exposure (i.e., step 1) were obtained from the "main singlet" of the spectra only. The UV-induced signals recorded from both vacuum- and non-vacuum-stored samples at RT with no light exposure were observed to have constant fading (up to 7 days) compared to the samples with light exposure. The sample sets with light exposure exhibited similar fading during the first few days (up to 3 days) followed by a subsequent signal increased at 7 days post-measurement. In contrast, a slower fading behavior of the UV-induced signal was observed in vacuum-stored samples at LT with no light exposure. This observation was slightly different from other studies, wherein fast signal decay was observed [1,12] in samples stored at LT, but in reasonable agreement with earlier observations in which ESR signals, in general, showed considerable stability [13,14].

The UV-induced signal (main singlet) from all the sample sets was found to have reduced its intensity immediately after AAT1 (data labeled as "soon after AAT1" in the plot) and then followed by signal growth. The behavior of the signal growth with time was also observed to be variable among different storage and location conditions and quite similar to that shown in Figure 4. While a larger increase in the signal intensity was seen in non-vacuum-stored AAT samples in comparison to the vacuum-stored, the AAT samples, in general, appeared to be sensitive to light. Thus, the most prominent way to keep the signal intensity moderately stable (or unchanged) was demonstrated by the vacuum storage at LT without light exposure. Furthermore, the intensity of the "bulge", which was also observed to be stable in vacuum-

stored samples, particularly at LT, instantly disappeared after AAT1 (spectral data not shown). The AAT also cut the chance of recurrence of the 'bulge' in all samples stored under different conditions (up to 7 days). This observation demonstrated that the application of AAT was able to sufficiently remove the signal contribution from heat-related effects (i.e., bulge) and reduce the intensity of the UV-induced signal (i.e., main singlet).

Next, the X-ray irradiation of the samples was conducted 7 days after AAT1. The sensitivity to radiation of the AAT samples was observed to be invariant (exhibited a fairly constant dose-response) with the storage conditions (i.e., vacuum storage at RT, non-vacuum storage at RT, and vacuum storage at LT). The radiation sensitivity and fading behavior of the vacuum-stored samples at RT, in both with and without light exposure, were comparable to each other, and the same relationship was also observed for those in non-vacuum-stored. The dose–response of the vacuum-stored samples at RT was higher by a factor of 1.5–2 compared to non-vacuum-stored (also at RT), and about 3 times lower than that of vacuum-stored samples at LT. An interesting observation in this plot was the similarity between the initial dose-response to 20 Gy of the vacuum-stored samples at LT without light exposure in Experiment 2 (see Figure 7E) and the initial dose-response of the AAT samples (before storage) in Experiment 1 (see Figure 4A–D). The premise behind the disparity of the radiation sensitivity may be related to the regrowth behavior of the UV-induced signal under various storage conditions following AAT. For instance, it was reported that the UV-induced signal can still be observed following water treatment when stored inside a tightly closed polyethylene bag under RT and LT conditions [1], although a rigorous comparison between the vacuum and non-vacuum storage (or whether the type of vacuum bags also matter) may be difficult to assess and requires further work.

Lastly, all five sample sets displayed a strong signal reduction following AAT2. The rate of change in the magnitude and stability of the signal was found to be higher in non-vacuum storage at RT with light exposure, suggesting that the light may have amplified the enhancement effect caused by the AAT. This observation was also consistent with the above results in Experiment 1, wherein the signal stability of the AAT samples depends strongly on storage conditions.

As a result of these observations, the AA was confirmed to be sensitive to light, demonstrating a substantial increase in the ESR peak-to-peak intensity to the fingernail spectrum. A permanent signal increase due to light sensitivity was also observed after repeated treatment. In our opinion, this variable contribution of light to the enhancement effect by the AA can be dealt with under well-controlled conditions. It can be concluded that the use of vacuum storage [2,15] at LT [13,14] with UV light restrictions [1,2] is recommended as a compromise for RIS fading but would also possibly suffer from equivalent drawbacks in keeping the unwanted signals intact (for example, UV-induced or heat-induced) which in turn could interfere with the determination of the RIS. In the current work, however, the scope of the study is very limited so far to 0.1 M concentration of the AA treatment solution. Further studies are required concerning the role of different concentration levels of AA as well as exploration of other antioxidants to better understand the complex ESR spectral characteristics of fingernails which could improve its potential as a dosimetric material.

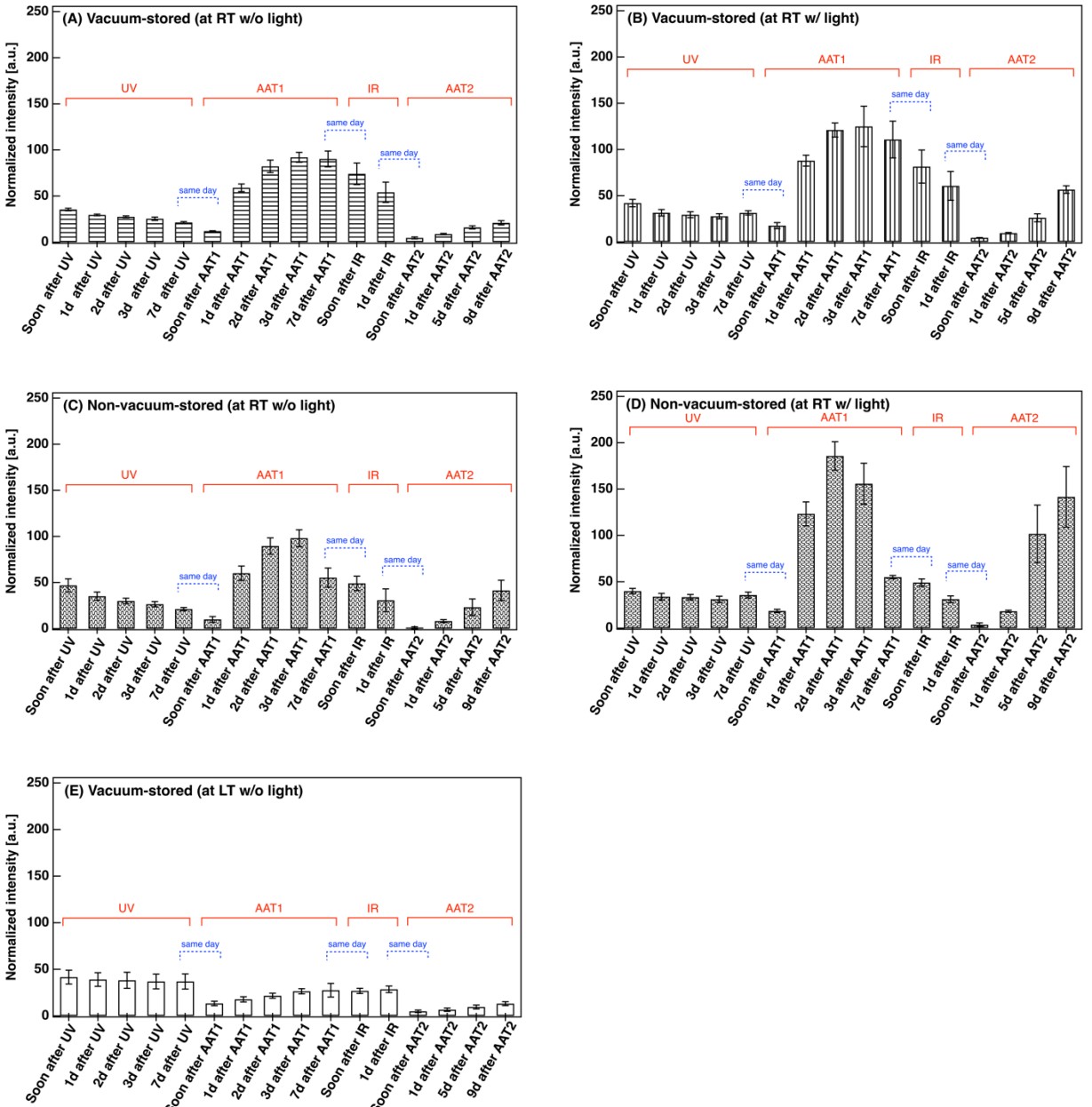

**Figure 7.** ESR peak-to-peak intensity changes obtained from fingernail samples stored under different storage conditions, as described in steps 1–4 of Experiment 2. Each point was calculated as the average of three AAT samples and subtracted from the WT control sample. The error bars were obtained as the standard deviation of three samples. The notations RT, LT, w/light, and w/o light are the same as in Figure 4.

## 4. Conclusions

The present study has attempted to investigate the effects of L(+)-ascorbic acid (AA) (a natural antioxidant commonly known as vitamin C) treatment on the electron spin resonance (ESR) signals of fingernails in relation to X-ray and UV irradiation. The spectral changes, including the magnitude and stability of the peak-to-peak intensities under various storage and location conditions, were also described. The results indicated that the signal stability of the irradiated and unirradiated samples treated with AA varied depending on the storage and location conditions. It was consistently observed that the AA treatment caused a dramatic increase in the ESR intensities and the continuous signal growth with time due to the sensitivity of the AA to light. As a positive finding, the AA

treatment effectively removed both the heat-related signal (i.e., bulge) and UV-induced signal (i.e., main singlet). Moreover, the increase in the ESR due to the AA treatment may provide an enhancement to the radiation-dependent signal of interest in fingernails. These findings are expected to improve the precision of radiation dosimetry using fingernails. Further studies, including theoretical research and experimental investigation with different antioxidant conditions, are needed for the verification of reproducibility and characterization of the complex changes of the ESR signals from fingernails.

**Author Contributions:** Conceptualization, C.A.B.G. and H.Y.; validation C.A.B.G. and H.Y.; investigation, C.A.B.G. and H.Y.; resources, H.Y.; data curation, C.A.B.G. and R.A.; writing—original draft preparation, C.A.B.G.; writing—review and editing, C.A.B.G. and H.Y.; supervision, H.Y.; funding acquisition, H.Y. All authors have read and agreed to the published version of the manuscript.

**Funding:** This work was supported in part by the Japan Society for the Promotion of Science (JSPS) KAKENHI under grant award 18KK0147; and also by the Program of the Network-type joint Usage/Research Center for Radiation Disaster Medical Science funded by the Ministry of Education, Culture, Sports, Science and Technology (MEXT) of Japan.

**Institutional Review Board Statement:** This study was approved by the Ethics Review Committee of Hiroshima University (Number E-1495).

**Informed Consent Statement:** Not applicable.

**Data Availability Statement:** Not applicable.

**Acknowledgments:** The authors are sincerely grateful to the editor(s) and peer reviewers for their constructive feedbacks and helpful comments. Chryzel Angelica B. Gonzales is also thankful to François Trompier from the Institute of Radiation and Nuclear Safety (IRSN) for his kind encouragement and guidance in the field of ESR dosimetry.

**Conflicts of Interest:** The authors declare no conflict of interest.

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
