# Peer review of "Experimental Study on the Effects of L(+)-Ascorbic Acid Treatment to the ESR Signals of Human Fingernails"

_applsci, doi:10.3390/app12168055_

Round 1

Reviewer 1 Report

The article is interesting, but the effect that ascorbic acid has on ESR and EPR needs to be more clear and more explicit, that is, an increase in the signal is indicative that it protects or simply that it is not having any effect.

In addition, it is important to consider that ascorbic acid is very unstable at temperature, humidity, light, and is even hydrolyzable, for this reason, it is important to discuss the results found considering these factors and relate them to your results to have a better understanding.

In general, article writing is good, but it is important that the results were written in the past since it is something that was done. Please review the specific comments in the attached file.

Reviewer 2 Report

Dear Colleagues, 

Dosimetry by measurement of intensity of EPR signals which are generated by radiation in living tissues is an important area of modern medical physics. Improving of the experimental approaches in this field is very useful work. Your numerous experiments seem to show the positive role of ascorbic acid in the study of EPR signals in the nails. Unfortunately, the presentation of the obtained results in the article is very unclear. The most important drawbacks are the following.

1. The essential observations are mixed with a lot of unimportant details. It is necessary to focus on the main results and to separate the secondary ones.

2. Large fragments of the text are simple descriptions of the figures. The figures discussion should contain generalizations and conclusions.

3. The multiple figures with EPR spectra which are similar to each other (Figures 6, 7, 8) do not carry the useful information. However, it remains unclear whether the shape of the spectra changes under influence of the studied impacts. Usually, EPR spectra are compared by normalization on intensity or double integral.

4. An interesting observation is the increase in signal intensity after irradiation. It is not explained what is the reason of this effect.

5. English narration should to be significantly improved. 

In my opinion, the article should be completely reworked.

Round 2

Reviewer 2 Report

Dear Colleagues, 

The changes you made did not satisfy me at all. Currently, the text of the article is simply a listing of numerous measurements and a description in words of changes that can be seen in the figures. The following remains unclear:

1. Comparison of the shapes of EPR spectra in the samples subjected to various treatments. Are these spectra the same? Are the same radicals formed in the samples or are they different?

2. Change in the shape of the spectra during storage of samples under various conditions. Is it only the intensity of the signals that changes, or does the shape change as well?

3. Why are the results of the first experiment presented in the form of a convenient scheme that reflects the data of all experiments, while the results of the second experiment are presented in the form of a huge set of spectra?

4. Why do we need f-e drawings in figures 5,7,8,9, if all observed changes are reflected in f-figures?

5. What does “susceptibility of the AA agent to light” mean?

6. Please, show the article to a native English speaker. 
